# Retention of zirconia crown to oxidized titanium base abutment: Experimental research

**Rawikorn Krongvanitchayakul , Chatchai Kunavisarut \*

Department of Advanced General Dentistry, Faculty of Dentistry, Mahidol University, Bangkok, Thailand

\* Chatchai.kun@mahidol.ac.th

**Data Availability Statement:** All relevant data are within the paper and its Supporting Information files.

## Abstract

### Statement of problem

To separate the crown from the titanium base abutment, by using heat, caused oxidization of the titanium base abutment. The effect of this procedure on the retention of a crown is unclear.

### Purpose

To compare the resin bond strength and failure type between zirconia crowns and titanium base abutments utilizing four different surface treatments. Surface roughness and morphology of each surface treatment were also investigated.

### Material and methods

Forty titanium base abutments (Variobase®) were divided into four groups, 1. Control, 2. Air abraded, 3. Oxidized, and 4. Oxidized-air abraded. Oxidized and oxidized-air abraded groups were debonded from zirconia crowns using constant dry heat at 500 ˚C. For air abraded and oxidized-air abraded groups (after oxidization), the titanium base abutments were air abraded with $Al_3O_2$. After cleaning, one specimen of each group was investigated under a non-contact profilometer (50x), then the same samples were investigated under SEM at 25,300,500,1000 magnification and EDS at 30kV of accelerated voltage. All specimens were then cemented (RelyX Ultimate). After aging, with thermocycling under 5C˚ to 55C˚,120 seconds dwell time for 5,000 cycles, bond strength was tested and statistical differences were calculated with One-way ANOVA (p-value <0.05) follow by Tukey test. All separated crowns and titanium base abutments were investigated under a light microscope (20x), using fisher's exact test for correlation of the failure types.

### Results

There was a significant difference in the mean value of tensile bond strength among the control and test groups. Comparisons between control(237.6±46.3N) and oxidized(241.7±46.3N) showed statistically different values from air abraded(372.9±113.2N) when assembled using different surface treatments of the titanium-based abutments. (p-value<0.005)

**Funding:** The authors received no specific funding for this work.

**Competing interests:** The authors have declared that no competing interests exist.

As for failure type, there were statistically significant differences between control versus air abraded, control versus oxidized-air abraded, oxidized versus air abraded, and oxidized versus oxidized-air abraded. (p-value<0.001) The titanium surface morphology shown from the profilometer and SEM was coordinated. Control (Ra 333.8nm) and oxidized (Ra 321.0nm) groups surfaces showed smooth, corrugated surfaces, meanwhile air abraded (Ra 476.0nm) and oxidized-air abraded (Ra 423.8nm) groups showed rough, rugged surfaces.

## Conclusion

Heat oxidization of titanium-based abutments did not adversely affect tensile bond strength or the failure mode and surface roughness between titanium base abutments and zirconia crowns. However, air abrasion of the titanium surface increased surface roughness and retentive strength.

## Clinical implications

The titanium base abutments that were oxidized under heat treatment did not have an effect on crown retention. Thoroughly air abraded the titanium abutment prior to cementation can increase cement bond strength.

## Introduction

Nowadays, a cemented crown on a titanium base abutment is widely used for implant restoration. This method has several distinct advantages that are passive fit, esthetic, stability, biocompatibility, and retrievability of being used for both cement and screw retained restorations [1–3].

After years of usage, restorations on implants may need to be repaired. Studies have reported that more than 10 percent of restorations on single dental implants were broken or chipped after one year [4, 5]. Additionally, an interproximal space occurred up to 66 percent of the time if the dental implant was placed next to a natural tooth [6]. These cases necessitate either reparation or a new restoration, depending on the severity of the damaged restoration [7]. With a screw retained restoration on a titanium base abutment, a dentist can easily remove and replace the restoration.

Dental laboratories can separate the crown from the titanium abutment using constant dry heat to break cement retention. A crown can be remade, then the separated abutment can be cleaned and recemented resulting in the desired restoration [8].

However, debonding the crown with a constant dry heat will cause oxidation on the abutment surface [9]. A study showed the oxidation kinetic of commercially pure titanium after dry heat and found that the titanium oxide layer ($TiO_2$) was altered [10]. The titanium oxide layer became thicker and rougher as the temperature increased [9]. When the temperature reached 276 ˚C crystalline phase began to occur and the abutment was crystallized entirely at 718 ˚C, creating a defect free surface [11]. The chemical bond between the phosphate group in resin cement and the titanium dioxide layer may be affected [10]. A study have shown that heating commercially pure titanium disk over 400 ˚C can decrease resin bond strength due to the thickening of the oxide layer [10]. However, a contrary study on the effects of debonding and rebonding different crown materials on $Ti_6Al_4V$ disks, found that heating titanium to 320 ˚C for 2 minutes did not affect the zirconia bond using resin cement but did result in an increased bond strength of lithium disilicate [8].

Previous studies on the effect of heat oxidation titanium are focused on shear bond strength, which may not represent actual clinical situations. Crown retention depends on multiple factors such as abutment morphology [12], cement type [13], cement gap [14], and surface treatment [15–17]. There has been no study for crown retention in heat oxidized titanium abutments that simulated clinical situations.

The purpose of this study was to compare the retention of zirconia crowns between the use of new titanium abutments and reusing heat debonded titanium abutments, with both air abraded and non-air abraded. The null hypothesis was that heat oxidation and air abrasion of titanium abutment would not affect the bond strength of zirconia crown using resin cement. The second null hypothesis was there is no correlation between failure patterns and abutment surface treatments.

## Material and methods

To estimate the sample size for this study calculation for 4 independent means will be used. The sample size was calculated with G-power 3.1.9.4 program with Alpha 0.05 and Power at 90 using mean and standard deviation from the previous study [14]. The sample size calculated is 10 pieces per group.

Implant analogs were embodied with epoxy resin into PVC blocks with diameters of 15 mm and 30 mm in height. A vertical distance of 3 mm was left from the implant shoulder to simulate vertical bone loss [15]. Titanium based abutments with a gingival height 1 mm, diameter 3.5 mm, and height of 5.5 mm in height (variobase®) were torqued to the analog at 35 Ncm. The scan body of the abutment was scanned by an intraoral scanner (3shape Trios®). Zirconia crowns were designed using a 3shape Dental system® software for the pull-off test by using a Universal testing machine (Instron® model 5566) as shown in the Fig 1. Luting space was set to 50 μm according to manufacturer's recommendation. Sixty zirconia crowns were milled from monolithic zirconia disk, then they were sintered and air abraded with $Al_2O_3$ 50 μm, 1 bar pressure, at 2 cm. distance.

Forty titanium base abutments were divided into 4 groups.

Group1: Unconditioned titanium base abutment, as a control group

Group2: Air abraded titanium base abutment. All specimens were air abraded with $Al_2O_3$ 50 μm with a pressure of 1 bar for 10 seconds. Abutments were fixed with customized rotation grip (Fig 2) which placed the abutment at 10mm distance and 45 degrees from air abrasion tip. The motor rotated at 12 rpm to ensure complete air abrasion of the abutment's surface area [15].

Group3: Oxidized titanium base abutments. The titanium base abutments were bonded with zirconia crowns using Relyx™ Ultimate and left on a bench for 24 hours, then the crowns were debonded with constant dry heat at 500 ˚C for 5 minutes. This was achieved by increasing the heat rate in increments of 45 ˚C using a porcelain furnace.

Group4: Oxidized and air abraded titanium abutment. The same bonding and debonding procedures as group 3 were used, in addition the specimens were air abraded under the same air abrasion protocol as group 2.

After the abutment's surface was treated and cleaned, one abutment from each of the four groups was selected at random to examine the surface roughness using a non-contact profilometer. An average surface roughness (Ra), mean height of the profile (Rc) and maximum height of the profile (Rz) were extracted from a single profile (2x2 mm$^2$) of the random abutment under 50x magnification. Then abutment samples were covered with a thin layer of gold (SC7620 Mini Sputter Coater; Quorum technologies, Ashford, UK) to investigate surface changes under scanning electron microscope (SEM)(JSM-6610LV; JEOL, Tokyo, Japan) at

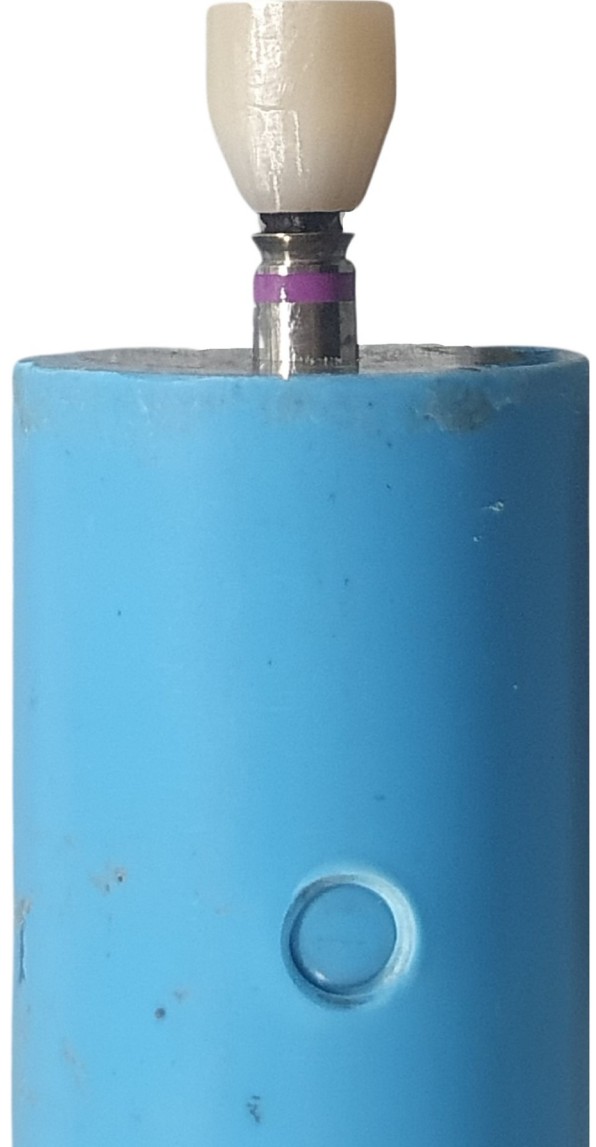

**Fig 1. Zirconia crown design for pull-out test using a Universal testing machine.**

300, 500 and 1000 magnification. Simultaneously, the same abutments were investigated composition under energy dispersion X-ray (EDS)(X-Max, Oxford, UK) at 30 kV of accelerated voltage [17].

Prior to bonding, all titanium base abutments and zirconia crowns in each of the four groups were cleaned using an ultrasonic bath and soaked in 99.5 percent acetone for 5 minutes and then dried with an air blower. Then a phosphate bonding (Scotchbond™ Universal) was applied for 20 seconds, and air blasted to remove the excess bonding before cementation. The specimens were cemented by one operator using firm finger pressure for 5 seconds then carefully cleaned of any excess cement. An additional 10 kg of pressure was applied for 6 minutes until the cement was fully set.

All specimens were left at room temperature for 24 hours before stimulating the aging process by using thermocycling at 5 ˚C to 55 ˚C total of 5,000 thermal cycles and dwell time of 120

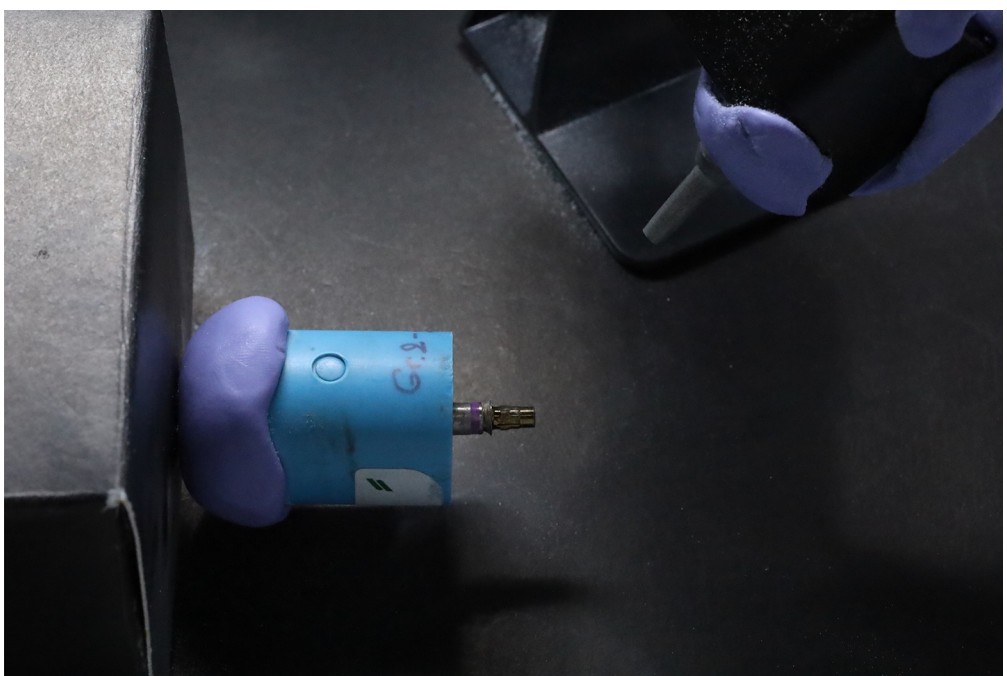

**Fig 2. Customized rotation grip for air abrasion protocol.** Abutment was place at 10mm. distance and 45 degrees from air abrasion tip. The motor rotation set at 12 rpm.

seconds [17]. A retention test was performed using a Universal testing machine INSTRON 5566, under 1-kN load cell, with a pullout speed of 1mm/min [4].After completion of the retention force test, the unbound crowns and abutments were inspected under 20x magnification light microscope for residual resin cement in order to accurately categorized the failure mode. Failure modes are categorized into 3 types:Type 1: Cement remain on the abutment surface > 90 percent;

Type 2: Cement remain on both side between 10–90 percent;

Type 3: Cement remain on crown > 90 percent (Fig 3).

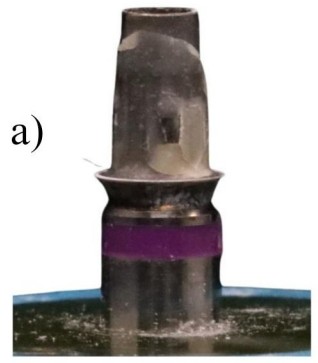 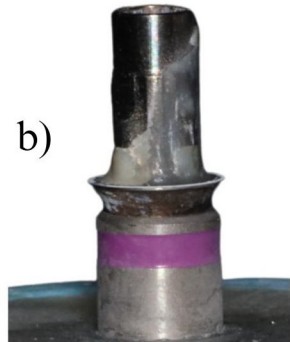 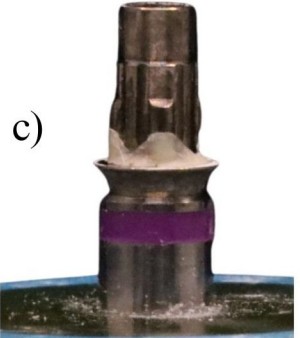

**Fig 3. Failure mode type.** a) Titanium base abutment show failure mode Type1, cement remained on the abutment surface >90%, b) Titanium base abutment show failure mode Type2, cement remain on the abutment between 10–90%; c) Titanium base abutment show failure mode Type3, cement remain on the abutment <10%.

**Table 1. Tensile bond strength(N) and failure mode of the control, air abraded, oxidized, and oxidized-air abraded titanium base abutment groups to zirconia crown using resin cement.** Failure mode investigated under light microscope(x20) (Type1, cement remained on the abutment surface >90%; Type2, cement remain on both side between 10–90%; Type3, cement remain on abutment <10%).

| | Groups | | | | p-value |
|---|---|---|---|---|---|
| | **Control (n = 10)** | **Air abraded (n = 10)** | **Oxidized (n = 10)** | **Oxidized-air abraded (n = 10)** | |
| Bond strength(N) | 237.6 ± 46.3 | 372.9 ± 113.2 | 241.7 ± 74.6 | 297.0 ± 76.3 | <0.005[a] |
| Failure mode n (%) | | | | | <0.001[b] |
| Type 1 | 0 | 10(100) | 0 | 7(70) | |
| Type 2 | 0 | 0 | 1(10) | 3(30) | |
| Type 3 | 10 (100) | 0 | 9 (90) | 0 | |

a Statistically significant differences from bond strength (One way-ANOVA)

b Statistically significant differences from failure mode (fisher's exact test, Chi-square)

For mean tensile bond strength and failure mode, statistic estimation of data distribution and comparison was performed using Windows-compatible SPSSTM Statistics version 20, IBM® Corporation, USA. The normal distribution and the variances of the data were tested. Tensile bond strength has normal distribution and variances of the data were equal, One-way ANOVA was used to calculate the possibility. And the significant differences between groups and multiple comparisons was tested with the Tukey test. Mean, SD, and 95% confidential interval were presented. As for failure mode, the non-parametric fisher exact test is performed. Surface roughness was shown as Median(Min,Max) along with a descriptive analysis.

## Results

There was a significant difference in the mean value of tensile bond strength among the control, air abraded, oxidized, and oxidized-air abraded groups. The mean tensile bond strength of the control group was 237.6 ± 46.3 N, the oxidized group was 241.7 ± 74.6 N, the air abraded group was 372.9 ± 113.2 N and the oxidized-air abraded group was 297.0 ± 76.3 N (Table 1, Fig 4). Control and oxidized groups appeared to be statistically and significantly different from

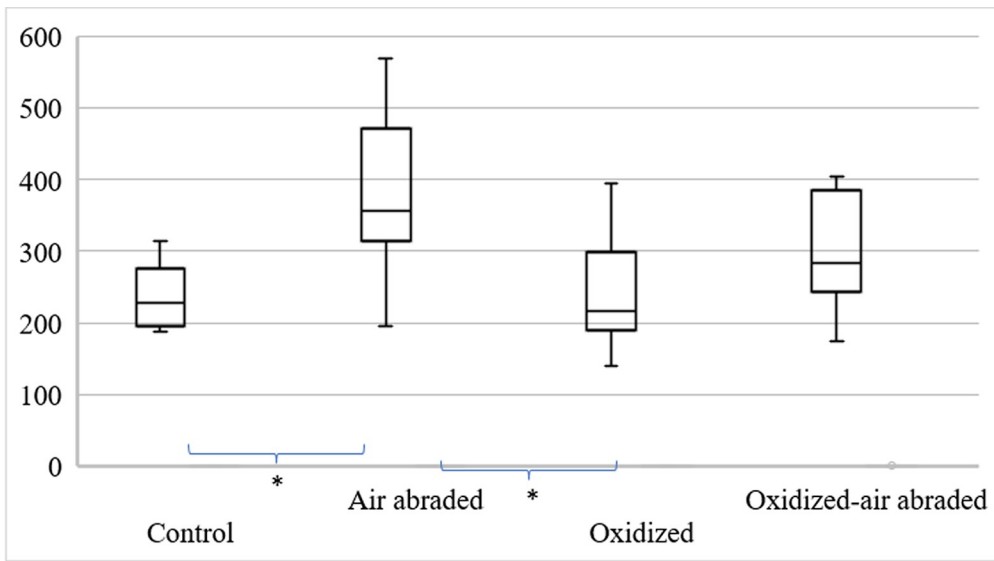

**Fig 4. Tensile bond strength(N) of control, air abraded, oxidized and oxidized-air abraded titanium base abutment to zirconia crown using resin cement.** * P-value <0.005(One way-ANOVA,Tukey test).

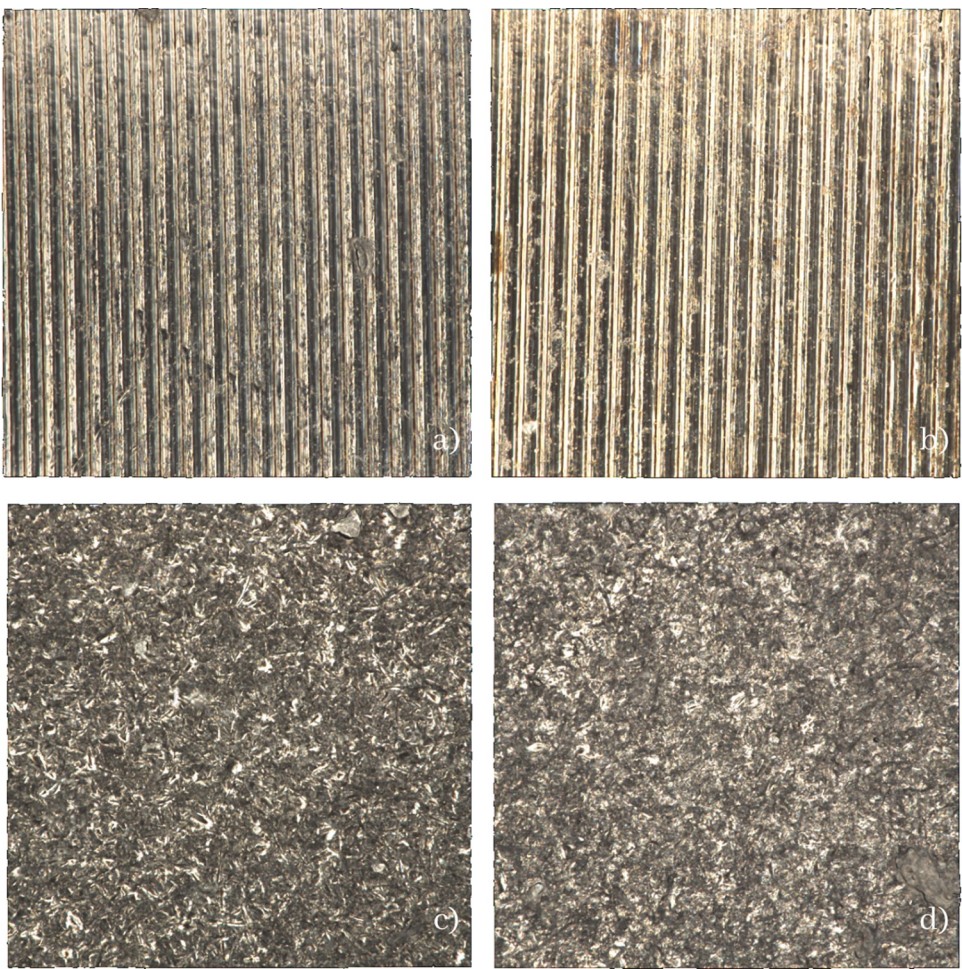

**Fig 5. Surface of titanium base abutment investigated under non-contact profilometer Alicona (x50).** a) Control-normal titanium base b) Air abraded titanium base c) Oxidized titanium base d) Oxidized-air abraded titanium base.

the air abraded group (p < 0.005). However, there was no significant difference between the oxidized-air abraded group and the air abraded group.

As for the surface morphology, despite the color, the control and oxidized groups had a uniform corrugated surface, while the air abraded and oxidized-air abraded groups showed a rough, rugged surface (Fig 5). Air abraded group have the highest surface roughness(Ra 476nm) follow by oxidized air abraded group(Ra 423.8nm),control group(Ra 333.8nm) and oxidized group(Ra 321nm), respectively (Table 2).

Similarly to the investigation from SEM, both control (Fig 6) and oxidized (Fig 7) groups showed a uniform corrugated surface, while the air abraded (Fig 8) and oxidized-air abraded

**Table 2. Median(Min, Max) value of titanium base abutment surface roughness investigated under non-contact profilometer Alicona (x50), extracted 3 times from a single profile photo.**

| | Groups | | | |
|---|---|---|---|---|
| | **Control** | **Air abraded** | **Oxidized** | **Oxidized-air abraded** |
| Ra(nm) | 333.8 (333.7, 379.7) | 476.0 (470.6, 579.3) | 321.0 (314.6, 363.7) | 423.8 (408.3, 440.8) |
| Rc(nm) | 419.6 (403.6, 488.5) | 607.9 (577.2, 995.1) | 402.7 (395.4, 484.9) | 555.9 (546.1, 605.9) |
| Rz(μ) | 2.23 (1.78, 2.54) | 3.21 (2.70,4,76) | 1.99 (1.89, 3.35) | 2.95(2.93,3.14) |

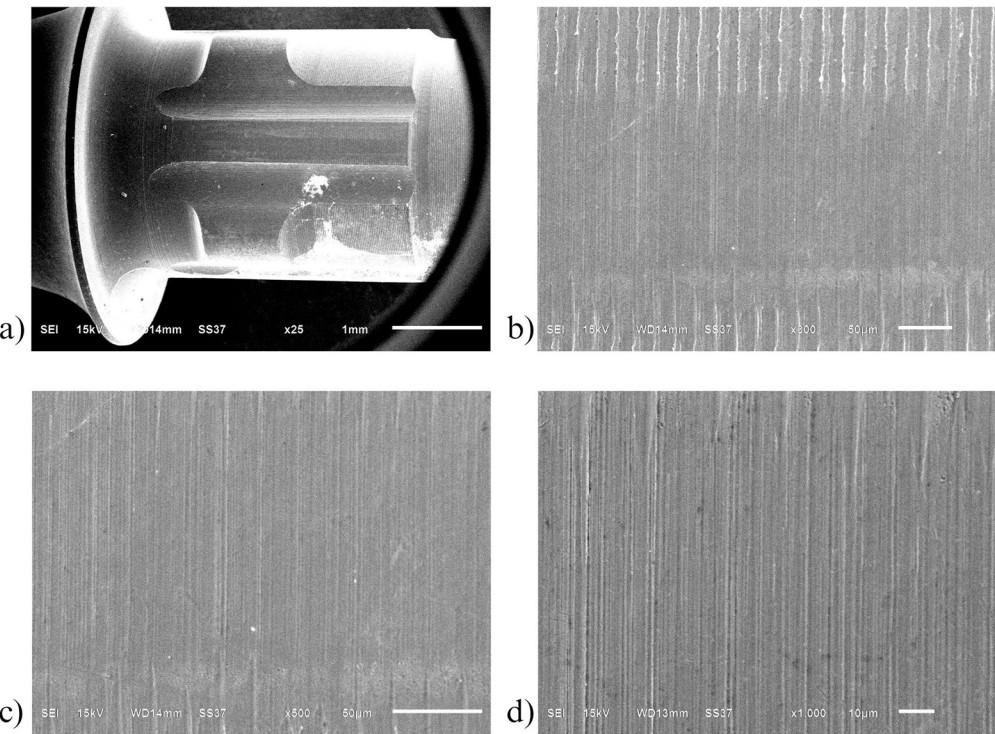

**Fig 6. Surface of non treated titanium base abutment(control) investigated under Scanning electron microscope (SEM).** a) 25 magnification b) 300 magnification c) 500 magnification d) 1000 magnification.

(Fig 9) groups showed a rugged surface. But the oxidized abutment surface showed multiple white dirt. EDS investigation reviewed only the oxidized group that presented calcium element on the surface, this reaffirmed the possibility that the white dirt was residual cement (Fig 10). There were differences in element both weight and atomic percent between control VS oxidized and control VS air abraded, oxidized-air abraded (Table 3). In addition, at 25 magnification, a cracked line was presented on the abutment surface of both air abraded and oxidized-air abraded groups (Figs 11–13).

The separate titanium base abutment and zirconia crowns were inspected under a 20x light microscope to determine the failure type. Control and oxidized groups presented only type 3 failure mode. While air abraded and oxidized-air abraded groups presented primarily type 1 failure mode. Test results showed significant statistical differences between the four groups. ($p < 0.001$). Therefore, the second null hypothesis was denied. There were statistically significant differences between (control vs. air abraded), (control vs. oxidized-air abraded), (oxidized vs. air abraded), and (oxidized vs. oxidized-air abraded) groups. (p-value $< 0.001$) (Table 1).

## Discussion

This study focused on the effect on the resin bond strength of heat oxidized titanium abutments under a strict debonding protocol using zirconia crowns. The results showed that heat oxidation of titanium base abutments did not have a statistically significant effect on resin bond strength. However, a notable difference in resin bond strength(p-value $< 0.005$) and failure mode(p-value $< 0.001$) was observed in the air abraded titanium group. Consequently, both hypotheses were rejected.

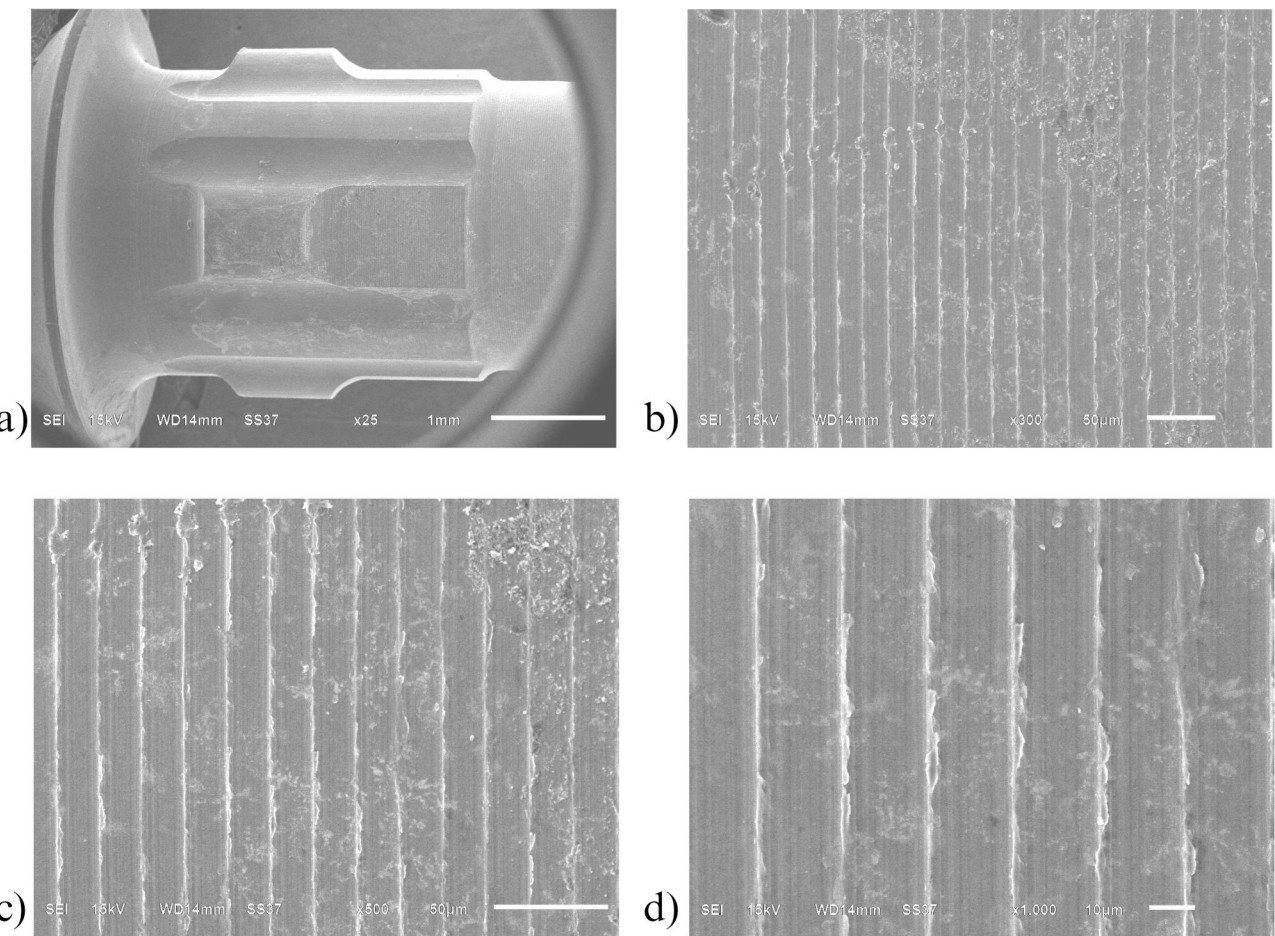

**Fig 7. Surface of oxidized titanium base abutment investigated under Scanning electron microscope (SEM).** a) 25 magnification b) 300 magnification c) 500 magnification d) 1000 magnification.

No measurable statistical differences in bond strength were found between the oxidized-air abraded group and the oxidized group, but a significant difference in the failure modes(p-value<0.001) was detected. The cement remained primarily on the "crown" in the oxidized group, whereas the opposite occurred for the oxidized-air abraded group, with the majority of the cement remaining on the "abutment." Air abrasion titanium abutments considerably increased the resin bond strength between the cement and abutment.

Resin cement is known to be bonded with titanium between the phosphate group and titanium oxide layer. The bonding depends on both micromechanical interlocking and physico-chemical state [18, 19]. The oxidization of titanium by using heat can change the physical and chemical properties of the oxide layer, which may affect resin bond strength.

This study found that heat oxidation of titanium at 500 ˚C for 5 minutes does not affect resin cement bond strength to zirconia, which is consistent with the previous study [8]. However, a study found that heating a cp titanium disk over 400˚C can decrease resin bond strength according to the thickening of the oxide layer [9]. This study attempts to imitate real clinical situations. As a result, the morphology of abutment [12], type of cement [13], and cement gap [14] can also be contributing factors for crown retention. There were no discernable differences observed in crown retention between the control and oxidized group in this study.

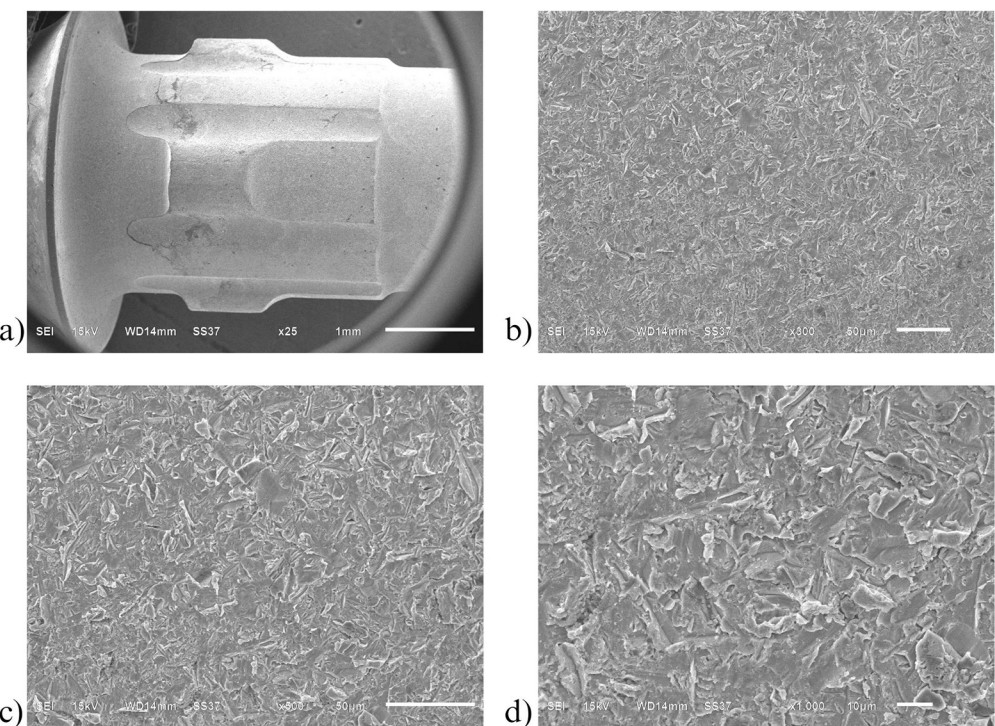

**Fig 8. Surface of air abraded titanium base abutment investigated under Scanning electron microscope (SEM).** a) 25 magnification b) 300 magnification c) 500 magnification d) 1000 magnification.

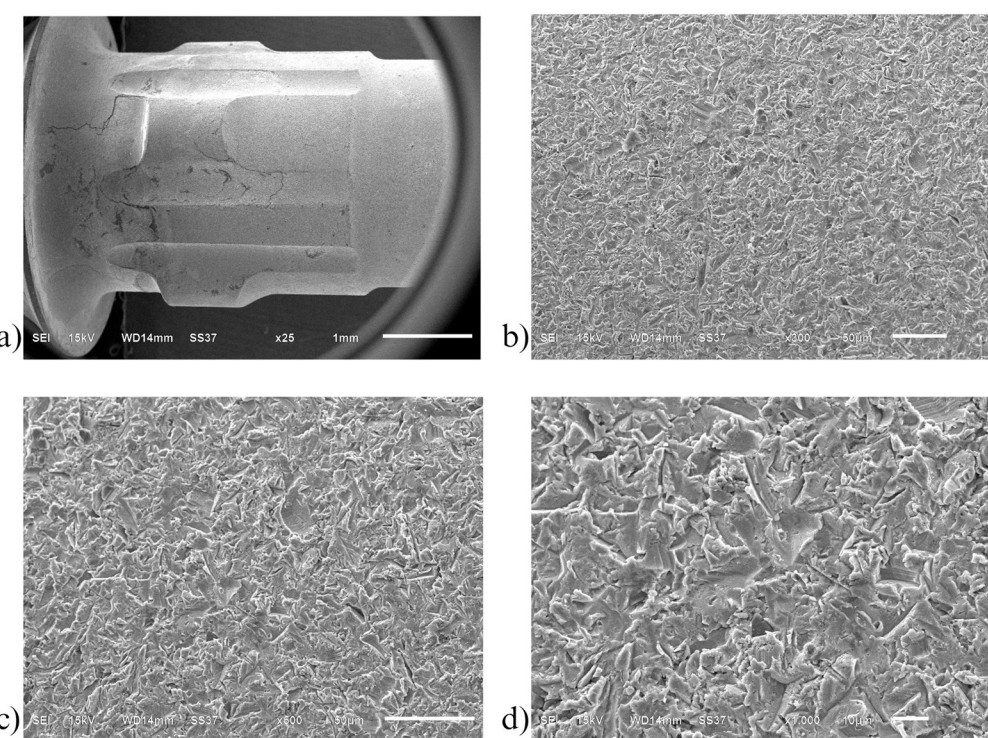

**Fig 9. Surface of oxidized-air abraded titanium base abutment investigated under Scanning electron microscope (SEM).** a) 25 magnification b) 300 magnification c) 500 magnification d) 1000 magnification.

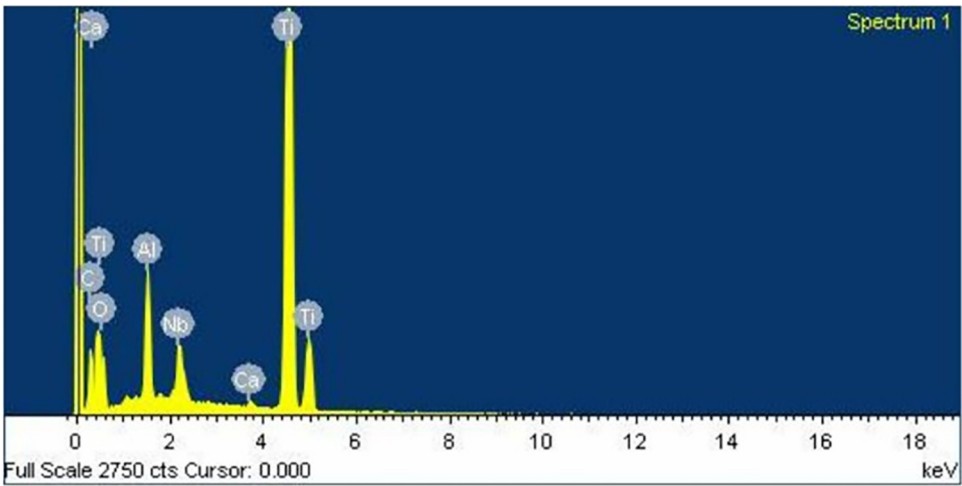

**Fig 10. Energy dispersion Xray (EDS) graph (X-Max,Oxford,UK) of a control-normal titanium base.**

**Table 3. Element composition of titanium base abutment surface investigated under energy dispersion Xray (EDS) (X-Max, Oxford, UK) at 30 kV of accelerated voltage.**

| Element | | Control | Air abraded | Oxidized | Oxidized- air abraded |
|---|---|---|---|---|---|
| C | Weight% | 12.88 | 8.09 | 9.13 | 8.09 |
|   | Atomic% | 31.65 | 20.16 | 23.56 | 20.16 |
| O | Weight% | 11.59 | 17.24 | 13.16 | 17.24 |
|   | Atomic% | 21.38 | 32.24 | 25.51 | 32.24 |
| Al | Weight% | 4.43 | 6.56 | 4.6 | 6.56 |
|   | Atomic% | 4.85 | 7.27 | 5.28 | 7.27 |
| Ca | Weight% | - | - | 0.34 | - |
|   | Atomic% | - | - | 0.27 | - |
| Ti | Weight% | 65.48 | 60.79 | 67.29 | 60.79 |
|   | Atomic% | 40.34 | 37.97 | 43.56 | 37.97 |
| Nb | Weight% | 5.62 | 7.32 | 5.48 | 7.32 |
|   | Atomic% | 1.78 | 2.36 | 1.83 | 2.36 |

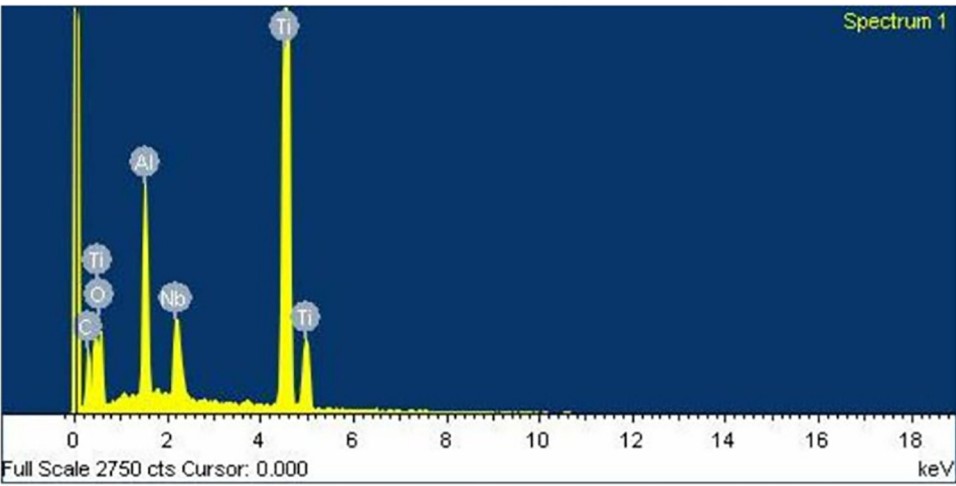

**Fig 11. Energy dispersion Xray (EDS) graph (X-Max,Oxford,UK) of an air abraded titanium base.**

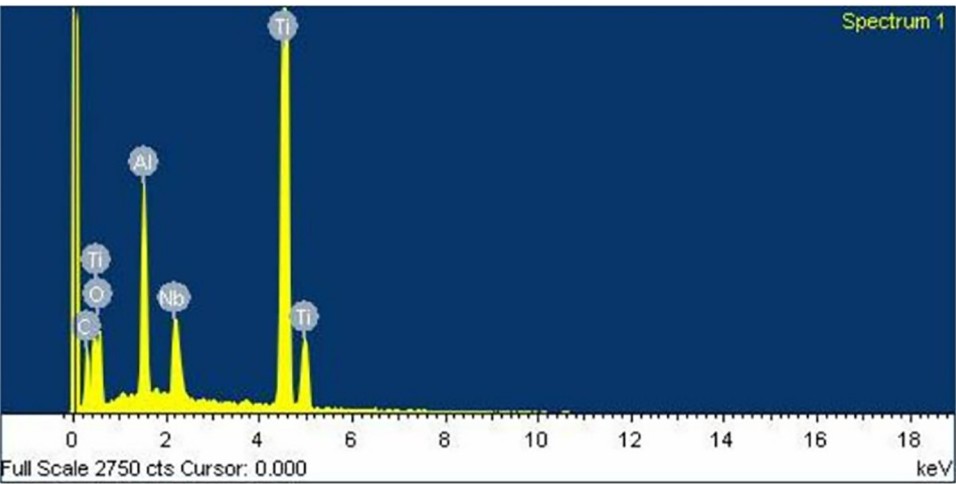

**Fig 12. Energy dispersion Xray (EDS) graph (X-Max,Oxford,UK) of an oxidized titanium base.**

Air abrasion the titanium abutments with $Al_3O_2$ 50 μm 2 bar pressure at a 10 mm distance for 10 seconds can increase crown retention and titanium surface roughness. Consistent to previous studies that air abrasion titanium greatly increases zirconia crown retention [15–17]. However, air abraded an oxidized titanium did not increase bond strength significantly when compared with both the controlled and oxidized groups. It was known that heat oxidation changes the oxide layer morphology and titanium's physical properties [10, 20]. From investigation under SEM and EDS, found no difference between air abraded and oxidized-air abraded in terms of surface morphology and elemental component. However, the titanium surface could be hardened when heating titanium due to titanium grain growth [21]. Supported by a study found that the low oxidation temperature of 450 ˚C followed by a high temperature of 850 ˚C for vacuum heat treatment caused a hard and defect-free surface of the titanium [22]. This study also found increasing of aluminum and oxygen elements after air abrasion. $Al_2O_3$ particles could be embedded in the metal surface effect to element component shift and increase resin bond strength [23].

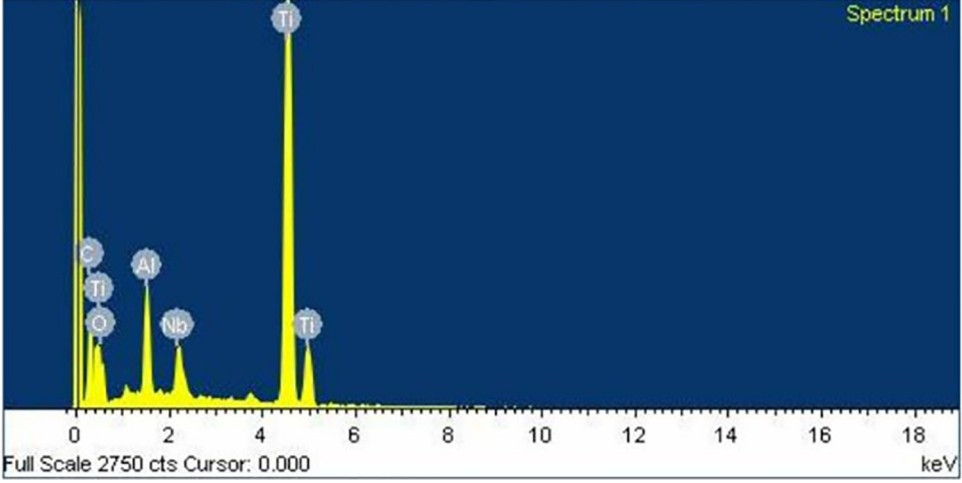

**Fig 13. Energy dispersion Xray (EDS) graph (X-Max,Oxford,UK) of an oxidized-air abraded titanium base.**

This study was conducted in a laboratory setting, which tries to imitate the clinical environment but may only partially reflect the conditions in a clinical setting, therefore, there should be clinical study to confirm this finding.

Based on results of this study, the oxidized titanium abutment can be used to bond with zirconia crown without compromising the retention between resin cement and restoration. However, air abrasion the titanium abutment did increase the bond strength. Therefore, air abrade the abutment is recommended before bonding restorations.

## Conclusions

The debonding protocol using heat to break resin bond did not affect zirconia crown retention using resin cement. Air abrade the titanium base abutment has an influence on both surface roughness and bond strength between abutment and zirconia restoration.

## Supporting information

**S1 Data.**
(XLSX)

**S2 Data.**
(XLSX)

## Author Contributions

**Conceptualization:** Chatchai Kunavisarut.

**Data curation:** Rawikorn Krongvanitchayakul, Chatchai Kunavisarut.

**Formal analysis:** Rawikorn Krongvanitchayakul, Chatchai Kunavisarut.

**Funding acquisition:** Chatchai Kunavisarut.

**Investigation:** Rawikorn Krongvanitchayakul, Chatchai Kunavisarut.

**Methodology:** Rawikorn Krongvanitchayakul, Chatchai Kunavisarut.

**Project administration:** Chatchai Kunavisarut.

**Resources:** Chatchai Kunavisarut.

**Supervision:** Chatchai Kunavisarut.

**Validation:** Chatchai Kunavisarut.

**Visualization:** Chatchai Kunavisarut.

**Writing – original draft:** Rawikorn Krongvanitchayakul.

**Writing – review & editing:** Chatchai Kunavisarut.

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
