## [Decision Letter · Decision Letter 0]

14 Feb 2023

PONE-D-22-35369Retention of zirconia crown to oxidized titanium base abutment: Experimental researchPLOS ONE

Dear Dr. Kunavisarut,

Thank you for submitting your manuscript to PLOS ONE. After careful consideration, we feel that it has merit but does not fully meet PLOS ONE’s publication criteria as it currently stands. Therefore, we invite you to submit a revised version of the manuscript that addresses the points raised during the review process.

Thank you for submitting the work to PLOS ONE. There are several concerns mainly on the manuscript writings, organization, and impact of the work. Please see the reviewers' comments for details. Please provide clarifications and address all comments.

We look forward to receiving your revised manuscript.

Kind regards,

Sompop Bencharit, DDS, MS, PhD, FACP

Academic Editor

PLOS ONE

Journal Requirements:

Additional Editor Comments:

Thank you for submitting the work to PLOS ONE. There are several concerns mainly on the manuscript writings, organization, and impact of the work. Please see the reviewers' comments for details. Please provide clarifications and address all comments.

Reviewers' comments:

Reviewer's Responses to Questions

**Comments to the Author**

1. Is the manuscript technically sound, and do the data support the conclusions?

Reviewer #1: No

Reviewer #2: Partly

Reviewer #3: Yes

2. Has the statistical analysis been performed appropriately and rigorously? 

Reviewer #1: No

Reviewer #2: No

Reviewer #3: Yes

3. Have the authors made all data underlying the findings in their manuscript fully available?

Reviewer #1: Yes

Reviewer #2: No

Reviewer #3: No

4. Is the manuscript presented in an intelligible fashion and written in standard English?

Reviewer #1: No

Reviewer #2: Yes

Reviewer #3: Yes

5. Review Comments to the Author

Reviewer #1: After reviewing the article, I point out the following topics for the authors to evaluate:

1. On page 11, in the last paragraph of the introduction, the authors point out a single null hypothesis. However, on page 17, they state that the second null hypothesis was confirmed.

2. On page 13, the authors state that a 2X2mm2 area of a single specimen from each group was used to assess roughness. However, on page 16, in table 1, they state that roughness was measured in 3 different specimens (n=3).

3. On page 14, figure 3, the authors describe 6 different situations to characterize 3 types of failures. The descriptions of the situations in the letters D, E, F, are difficult to visualize and do not add relevant information to the situations described in the letters A, B, C. Therefore, I suggest keeping only the characterizations of the letters A, B, C to describe the 3 types of failure.

4. The main limitations of the article presented are the absence of a sample calculation, or the calculation of the power of the sample used, and also the lack of description of the statistical analysis performed.

5. In the results, on page 15, I also suggest reviewing the mean and standard deviation values of one of the roughness parameters (Rc). Did the sandblasted and oxidized sandblasted groups show exactly the same values?

6. On page 15, in Graph 1, I suggest adding indicators whether or not there was a statistically significant difference between the groups.

7. On page 16, table 1, a single p value is presented for two different comparisons of the Ra parameter, described in the legend (control vs. sandblasted) and (sandblasted vs. oxidized).

8. On page 16, still in Table 1, a single p value is presented for four different comparisons of the Rc parameter, described in the legend (control vs. sandblasted), (control vs. oxidized-sandblasted), (Oxidized vs. sandblasted) and (oxidized vs. oxidized-sandblasted).

9. On page 16, in the presentation of the groups, I suggest keeping the sandblasted group as letter b, to correspond to group 2, which was always presented as sandblasted.

10. On page 17, in table 2, it is described that the evaluation of the type of failure was performed with a magnification of X10. In other locations, this described magnification was X20.

11. Still on page 17, in table 2, I suggest reviewing the data from the sandblasted, control and oxidized-sandblasted groups. As the sandblasted group had the highest roughness, wouldn't it be expected that the sandblasted groups would have more type 1 failures? On the other hand, the control and oxidized-sandblasted groups, with less roughness compared to the sandblasted group, wouldn't more type 3 failures be expected?

12. Still on page 17, in the caption of table 2, I suggest repeating the description of the types of failures to facilitate the reading and understanding of the text.

13. In the discussion, on page 18, line 8, wouldn't the comparison described be between oxidized sandblasted group and sandblasted group rather than oxidized sandblasted group and oxidized group?

14. Still on page 18, the authors state: “The majority of the cement remaining on “crown” in oxidized group while the opposite occurred for the oxidized sandblasted group, with the majority of the cement remaining on the “abutment”. Table 2 does not demonstrate this. Note that in the oxidized group, most failures were type 1, which 90% of the cement remained retained in the abutment. As described in item 11, I suggest reviewing the data presented in table 2.

15. On page 19, lines 12 and 13, the authors state: “Sandblasting protocol in this study may not cause enough roughness to increase bond strength significantly.” In the results presented, blasting resulted in increased retention when compared to the control group, which contradicts this statement.

16. It would be important for the authors to point out the possible limitations of their work. As an example, the lack of physicochemical characterization of Ti surfaces before and after oxidation. Furthermore, the method used to characterize the abutment surface in the 4 different groups seems very limited. Perhaps it would be interesting to use other resources such as SEM.

Reviewer #2: Main concern:

The present study has a small impact on the dental field. Most of the cited studies on the subject are old, reinforcing the small impact of the subject in the current context. In addition, several points in the writing of the manuscript must be revised and reformulated to be in Plos One standards. The study did not adequately characterize the problem in the introduction, did not present statistical analysis methods, and did not discuss limitations. In addition, several references presented in the text are not in the references cited at the end of the text.

Abstracts.

- The first sentence of the abstract is strange. Please, review it.

- Explain briefly the method of specimen aging.

- In the results, the authors reported the statistical difference between the groups. However, it is important to present the directions of the effects, i.g., which group was better?

- Please, present the mean and SD of the tests results.

Introduction

- Please, avoid direct citations in the text.

- The sentence “debonding the crown with a constant dry heat will cause oxidation on the

- abutment surface.” needs to be referenced.

- The affirmation of the sentence “These cases provide the need for a new restoration.” Including chipped restoration can lead to an overtreatmnent. Restoration repair should be the first option in these cases. So, I suggest to the authors the reading of below paper:

Anusavice KJ. Standardizing failure, success, and survival decisions in clinical studies of ceramic and metal-ceramic fixed dental prostheses. Dent Mater. 2012;28:102-111

- The authors report the study of Gamelli E et al. However, this study is not in the reference

- The reference cited in the introduction Sailor et al is not cited in the reference

- The sentence “The chemical bond between the phosphate group in resin cement and the titanium dioxide layer may be affected.” need to be referenced. Also, explain how the layer may be affected.

- The authors need to improve the introduction to better characterize the problem of the study. How the oxide affect the cementation? In which clinical situations the dry heat debondation is indicated?

Materials and methods

- The authors report “Sixty zirconia crowns were milled from monolithic zirconia disk”. However, seems that there are ten “n” for the group. Please, clarify this in the study.

- The authors did not report how the statistic was performed in the study.

Results

- Show where the statistical differences are in graph 1. Also, why the authors used a bo plot graph? Box plot should be used for nonparametric distribution. Did the authors check the data distribution? what test was used?

- Table 1. Please, provide similar letters for groups that were similar in the analyzes and different letters for different groups. This is the conventional way of reporting statistical differences in this type of study.

- Present each statistical test performed in the footnote of the table/figure.

- Avoid rewriting all the data (already presented in the tables) in the text. Draw attention to what really matters in the text.

Discussion

- Authors need to discuss the limitations of the present study and the directions for further studies.

- Avoid direct citations.

References:

- Please, avoid the citations of books and old references

- Some references are not under the journal style

Reviewer #3: The study aimed to evaluate potential surface treatments for debonded crown on titanium base abutment. The results are interesting and justified by the methods used. It would be interesting to add in the discussion the explanation of how the debonded method used in the paper can relate to clinical condition, and also the clinical application of sandblasting, as if it is safe to perfome it clinically.

6. PLOS authors have the option to publish the peer review history of their article (what does this mean?). If published, this will include your full peer review and any attached files.

Reviewer #1: No

Reviewer #2: **Yes: **Luiz Alexandre Chisini

Reviewer #3: No

---

## [Author Response · Author response to Decision Letter 0]

30 Mar 2023

Thank you for taking the time to review my manuscript. I appreciate your feedback; they were accommodating. I have incorporated all of your suggestions in my revisitation as described in this letter.

Reviewer #1:

1. On page 11, in the last paragraph of the introduction, the authors point out a single null hypothesis. However, on page 17, they state that the second null hypothesis was confirmed.

Answer: The second hypothesis was added to the introduction.

2. On page 13, the authors state that a 2X2mm2 area of a single specimen from each group was used to assess roughness. However, on page 16, in table 1, they state that roughness was measured in 3 different specimens (n=3).

Answer: The roughness was only measured in a single specimen from each group. Table1 has been corrected accordingly.

3. On page 14, figure 3, the authors describe 6 different situations to characterize 3 types of failures. The descriptions of the situations in the letters D, E, F, are difficult to visualize and do not add relevant information to the situations described in the letters A, B, C. Therefore, I suggest keeping only the characterizations of the letters A, B, C to describe the 3 types of failure.

Answer: Figure3 was corrected to remain only cement retained on titanium base abutment as letter A,B,C. Letter D,E,F as figure of cement retained on crown has been removed.

4. The main limitations of the article presented are the absence of a sample calculation, or the calculation of the power of the sample used, and also the lack of description of the statistical analysis performed.

Answer: As for sample size calculation, I add the explanation in the first paragraph of the material and method(page4line238-332). And for description of the statistical analysis, I have explained the statistical analysis in the last paragraph of material and method (page 9, lines 444-453) 

5. In the results, on page 15, I also suggest reviewing the mean and standard deviation values of one of the roughness parameters (Rc). Did the sandblasted and oxidized sandblasted groups show exactly the same values?

Answer: The surface roughness description in the paragraph has been removed and rephrased for better focus. And for better understanding Table2 has been modified to Median(Min,Max), according to the data type.

6. On page 15, in Graph 1, I suggest adding indicators whether or not there was a statistically significant difference between the groups.

Answer: The indicators was added to the graph.

7. On page 16, table 1, a single p value is presented for two different comparisons of the Ra parameter, described in the legend (control vs. sandblasted) and (sandblasted vs. oxidized).

Answer: The statistic analysis of Ra has been removed, according its data type, which is a descriptive data extracted from one sample. Therefore, Table1 and 2 was modified accordingly and p-value of surface roughness was removed.

8. On page 16, still in Table 1, a single p value is presented for four different comparisons of the Rc parameter, described in the legend (control vs. sandblasted), (control vs. oxidized-sandblasted), (Oxidized vs. sandblasted) and (oxidized vs. oxidized-sandblasted).

Answer: The statistic analysis of Rc has been removed, according its data type, which is a descriptive data extracted from one sample. Therefore, Table1 and 2 was modified accordingly and p-value of surface roughness was removed.

9. On page 16, in the presentation of the groups, I suggest keeping the sandblasted group as letter b, to correspond to group 2, which was always presented as sandblasted.

Answer: The letter b in Figures legend and Tables was corrected to always presented as sandblasted. 

10. On page 17, in table 2, it is described that the evaluation of the type of failure was performed with a magnification of X10. In other locations, this described magnification was X20.

Answer: The table’s legend was corrected to x20 magnification.

11. Still on page 17, in table 2, I suggest reviewing the data from the sandblasted, control and oxidized-sandblasted groups. As the sandblasted group had the highest roughness, wouldn't it be expected that the sandblasted groups would have more type 1 failures? On the other hand, the control and oxidized-sandblasted groups, with less roughness compared to the sandblasted group, wouldn't more type 3 failures be expected?

Answer: The data in table was reviewed and corrected to be in the right order.

12. Still on page 17, in the caption of table 2, I suggest repeating the description of the types of failures to facilitate the reading and understanding of the text.

Answer: The failure type has been added to table 1, therefore caption of table1 has been added the types of failures mode description for more understanding.

13. In the discussion, on page 18, line 8, wouldn't the comparison described be between oxidized sandblasted group and sandblasted group rather than oxidized sandblasted group and oxidized group?

Answer: I intend to emphasize the differences found between two oxidized groups. I have rephrase the sentence to “However, a notable difference in resin bond strength(p-value<0.005) and failure mode(p-value<0.001) was observed in the sandblasted titanium group. Consequently, both hypotheses were rejected. It is worth mentioning that no measurable statistical differences in bond strength were found between the oxidized-sandblasted group and the oxidized group, but a significant difference in the failure modes(p-value<0.001) was detected.”(page14line777-779)

14. Still on page 18, the authors state: “The majority of the cement remaining on “crown” in oxidized group while the opposite occurred for the oxidized sandblasted group, with the majority of the cement remaining on the “abutment”. Table 2 does not demonstrate this. Note that in the oxidized group, most failures were type 1, which 90% of the cement remained retained in the abutment. As described in item 11, I suggest reviewing the data presented in table 2.

Answer: The failure type has been added to table 1. Though, the data of failure mode has been revised and corrected.

15. On page 19, lines 12 and 13, the authors state: “Sandblasting protocol in this study may not cause enough roughness to increase bond strength significantly.” In the results presented, blasting resulted in increased retention when compared to the control group, which contradicts this statement.

Answer: The phrase has been removed. The discussion was focusing on limitation and further investigation instead.

16. It would be important for the authors to point out the possible limitations of their work. As an example, the lack of physicochemical characterization of Ti surfaces before and after oxidation. Furthermore, the method used to characterize the abutment surface in the 4 different groups seems very limited. Perhaps it would be interesting to use other resources such as SEM.

Answer: The SEM along with EDS investigation was performed and added to this study. Moreover, I have discussed more point on limitations and suggestion for further investigation. 

Reviewer #2: 

Main concern:

The present study has a small impact on the dental field. Most of the cited studies on the subject are old, reinforcing the small impact of the subject in the current context. In addition, several points in the writing of the manuscript must be revised and reformulated to be in Plos One standards. The study did not adequately characterize the problem in the introduction, did not present statistical analysis methods, and did not discuss limitations. In addition, several references presented in the text are not in the references cited at the end of the text.

Answer: This study was performed for more understanding on whether we can reuse a titanium base abutment or not? Reusing the old abutment will save cost to the treatment. In fact, many clinicians are currently using the old abutments with the new restorations without knowing the consequences of oxidation on the abutments. 

While revising, I reformulated manuscript to Plos-one pattern, added the sample size calculation, statistical analysis, perform a SEM/EDS examination and discuss more on study limitation and further investigation. Moreover, I revised and avoid cite old references as possible. But some of my concerns, mostly on the study of metal metallurgy, are old, and there are no newer direct references on the issue. Such as “Assefpour-Dezfuly M, Vlachos C, Andrews E. Oxide morphology and adhesive bonding on titanium surfaces. Journal of materials science. 1984;19(11):3626-39.” And “Taira Y, Matsumura H, Yoshida K, Tanaka T, Atsuta M. Influence of surface oxidation of titanium on adhesion. Journal of dentistry. 1998;26(1):69-73.”.

Abstracts.

- The first sentence of the abstract is strange. Please, review it.

Answer: The first sentence was modified.

- Explain briefly the method of specimen aging.

Answer: The method of specimen aging has been added.

- In the results, the authors reported the statistical difference between the groups. However, it is important to present the directions of the effects, i.g., which group was better?

Answer: The mean and SD was presented to imply the direction of effects.

- Please, present the mean and SD of the tests results.

Answer: The mean and SD has been presented.

Introduction

- Please, avoid direct citations in the text.

Answer: The direct citation has been removed.

- The sentence “debonding the crown with a constant dry heat will cause oxidation on the

abutment surface.” needs to be referenced.

Answer: The phrase has been referenced.

- The affirmation of the sentence “These cases provide the need for a new restoration.” Including chipped restoration can lead to an overtreatmnent. Restoration repair should be the first option in these cases. So, I suggest to the authors the reading of below paper:

Anusavice KJ. Standardizing failure, success, and survival decisions in clinical studies of ceramic and metal-ceramic fixed dental prostheses. Dent Mater. 2012;28:102-111

Answer: I appreciated your suggestion, I have reviewed the journal and used it as reference.

- The authors report the study of Gamelli E et al. However, this study is not in the reference

Answer: The cited article has been added in the manuscript.

- The reference cited in the introduction Sailor et al is not cited in the reference

Answer: The cited article has been added in the manuscript.

- The sentence “The chemical bond between the phosphate group in resin cement and the titanium dioxide layer may be affected.” need to be referenced. Also, explain how the layer may be affected.

Answer: The reference for this statement has been added.

- The authors need to improve the introduction to better characterize the problem of the study. How the oxide affect the cementation? In which clinical situations the dry heat debondation is indicated?

Answer: I have rewrite my introduction for better understanding, and also indicated clinical situation which debonding is necessary. There is no previous study on the problem and there are only few related studies. 

Materials and methods

- The authors report “Sixty zirconia crowns were milled from monolithic zirconia disk”. However, seems that there are ten “n” for the group. Please, clarify this in the study.

Answer: To clarify the number of zirconia crowns used in this research, I phrase “Sixty zirconia crowns were milled from monolithic zirconia disk, then they were sintered and sandblasted with Al2O3 50 µm, 1 bar pressure, at 2 cm. distance. Twenty zirconia crowns were used to simulate debonding process on the test titanium-based abutments. The other forty zirconia crowns were used on the experiment.” in material and method (page 5, lines 336-339).

- The authors did not report how the statistic was performed in the study.

Answer: The statistical analysis has been added in the last paragraph of material and method (page 9, lines 444-453)

Results

- Show where the statistical differences are in graph 1. Also, why the authors used a bo plot graph? Box plot should be used for nonparametric distribution. Did the authors check the data distribution? what test was used?

Answer: The indicator of statistical differences was presented in graph1. The box-plot graph is presented for a better exhibition of both distribution of the data and the significant differences in groups, analyzed from the tukey test.

- Table 1. Please, provide similar letters for groups that were similar in the analyzes and different letters for different groups. This is the conventional way of reporting statistical differences in this type of study.

Answer: Table1 and 2 have been modified for better understanding and explicated the data type.

- Present each statistical test performed in the footnote of the table/figure.

Answer: The each statistical test performed in the footnote of the table/figure. 

- Avoid rewriting all the data (already presented in the tables) in the text. Draw attention to what really matters in the text.

Answer: The description of all data has been rewritten. Most of data which already presented in table was removed.

Discussion

- Authors need to discuss the limitations of the present study and the directions for further studies.

Answer: Limitation and direction for further studies have been discussed.

- Avoid direct citations.

Answer: The direct citation has been removed.

References:

- Please, avoid the citations of books and old references

Answer: Old references have been revised and some are remained. As the study of metal metallurgy, are old, and there are no newer direct references on the issue. Such as “Assefpour-Dezfuly M, Vlachos C, Andrews E. Oxide morphology and adhesive bonding on titanium surfaces. Journal of materials science. 1984;19(11):3626-39.” And “Taira Y, Matsumura H, Yoshida K, Tanaka T, Atsuta M. Influence of surface oxidation of titanium on adhesion. Journal of dentistry. 1998;26(1):69-73.”. 

- Some references are not under the journal style

Answer: The references are revised and rewritten.

Reviewer #3: The study aimed to evaluate potential surface treatments for debonded crown on titanium base abutment. The results are interesting and justified by the methods used. It would be interesting to add in the discussion the explanation of how the debonded method used in the paper can relate to clinical condition, and also the clinical application of sandblasting, as if it is safe to perfome it clinically.

Answer: The discussion has been rewritten and discuss more on limitation of the study, clinical application and direction for further studies.

Finally, I appreciate all your feedbacks. Thank you for your time and consideration.

---

## [Editor Report · Decision Letter 1]

14 Apr 2023

PONE-D-22-35369R1Retention of zirconia crown to oxidized titanium base abutment: Experimental researchPLOS ONE

Dear Dr. Kunavisarut,

Thank you for submitting your manuscript to PLOS ONE. After careful consideration, we feel that it has merit but does not fully meet PLOS ONE’s publication criteria as it currently stands. Therefore, we invite you to submit a revised version of the manuscript that addresses the points raised during the review process.

We look forward to receiving your revised manuscript.

Kind regards,

Sompop Bencharit, DDS, MS, PhD, FACP

Academic Editor

PLOS ONE

Journal Requirements:

Additional Editor Comments:

Thank you for addressing the comments from the reviewers. I think the manuscript is significantly improved. There are a few additional requests I would like to add.

1) Please add SEM method in the Methods section.

2) Since you compared the SEM analyses among different groups, I think we need to take it one step further to see if there is damages or changes in material elemental composition. Can you add EDX experiments into this? If not please provide justification and rationale. My thought is that air abrasion may alter the composition of the abutment surface.

3) Please change "sand blasting" to "air abrasion".

4) Can you discuss the depth of the air abrasion and the anodization layer? I assume that air abrasion would not go beyond the anodization layer. If we will recommend this clinically, would this be an issue if someone air abrade the implant abutment too much?

---

## [Author Response · Author response to Decision Letter 1]

17 May 2023

To the Editor and reviewers of the PLOS ONE journal

Dear Editor and reviewers, 

Thank you for taking the time to review my manuscript. I appreciate your feedback; they were accommodating. I have incorporated all of your suggestions in my revisitation as described in this letter.

1) Please add SEM method in the Methods section.

Answer: I have add SEM and EDS method in the Method section(page5 line 141-146)

2) Since you compared the SEM analyses among different groups, I think we need to take it one step further to see if there is damages or changes in material elemental composition. Can you add EDX experiments into this? If not please provide justification and rationale. My thought is that air abrasion may alter the composition of the abutment surface.

Answer: I have add the EDS analysis and add the result graph demonstrate material composition along with a table demonstrate element composition in weight% and atomic%. 

I also discuss finding from EDS on increasing of aluminum and oxygen after air abrasion. As reference “23. Ohkubo C, Watanabe I, Hosoi T, Okabe T. Shear bond strengths of polymethyl methacrylate to cast titanium and cobalt-chromium frameworks using five metal primers. The Journal of prosthetic dentistry. 2000 Jan 1;83(1):50-7.” After air abrasion, Al2O3 particles embedded into metal surface which increase resin bond strength.

3) Please change "sand blasting" to "air abrasion".

Answer: I have change all “sandblasting” to “air abrasion” including the name of groups “sandblasted group” to “air abraded group” and “Oxidized-sandblasted group” to “oxidized-air abraded group” for consistency of the manuscript.

4) Can you discuss the depth of the air abrasion and the anodization layer? I assume that air abrasion would not go beyond the anodization layer. If we will recommend this clinically, would this be an issue if someone air abrade the implant abutment too much?

Answer: Since this study did not investigate on the anodized abutment, we couldn’t make any assumption of this matter. Further research should be conducted to investigate the effect of air-abraded on anodized abutment.

Finally, I appreciate all your feedbacks. Thank you for your time and consideration.

Sincerely,

Assistant Professor Chatchai Kunavisarut

Business address: Department of Advanced General Dentistry, Mahidol University, 

No.6 Yothi Road, Ratchathewi District, Bangkok 10400 Hospital (Phayathai Campas) 

Tel:+662-200-7777 

Email: Chatchai.kun@mahidol.ac.th Mobile:+66890737030

---

## [Editor Report · Decision Letter 2]

30 May 2023

Retention of zirconia crown to oxidized titanium base abutment: Experimental research

PONE-D-22-35369R2

Dear Dr. Kunavisarut,

We’re pleased to inform you that your manuscript has been judged scientifically suitable for publication and will be formally accepted for publication once it meets all outstanding technical requirements.

Kind regards,

Sompop Bencharit, DDS, MS, PhD, FACP

Academic Editor

PLOS ONE

Additional Editor Comments (optional):

Thank you for revising this manuscript and addressing all comments from the reviewers and editor.